# LLMS ARE IN-CONTEXT REINFORCEMENT LEARNERS

## ABSTRACT

Large Language Models (LLMs) can learn new tasks through in-context supervised learning (i.e., ICL). This work studies if this ability extends to in-context reinforcement learning (ICRL), where models are not given gold labels in context, but only their past predictions and rewards. We show that a naive application of ICRL fails miserably, and identify the root cause as a fundamental deficiency at exploration, which leads to quick model degeneration. We propose an algorithm to address this deficiency by increasing test-time compute, as well as a compute-bound approximation. We use several challenging classification tasks to empirically show that our ICRL algorithms lead to effective learning from rewards alone, and analyze the characteristics of this ability and our methods. Overall, our results reveal remarkable ICRL abilities in LLMs.

## 1 INTRODUCTION

Large language models (LLMs) have been shown to exhibit in-context learning (ICL), a form of supervised learning that does not require parameter updates (Brown et al., 2020). ICL relies on including supervised input-output pairs in the LLM context (i.e., prompt),[1] and has been shown as effective with either few (Brown et al., 2020) or many (Bertsch et al., 2024; Agarwal et al., 2024) examples. In this paper, we ask whether the ability to learn in-context extends to the reinforcement learning (RL) paradigm, i.e., whether language models can effectively perform in-context reinforcement learning (ICRL).

ICRL is a natural combination of ICL and reinforcement learning (RL). Instead of constructing the LLM context from supervised input-output pairs, the LLM context is constructed using triplets

```
Query: It declined my transfer.
🦙 Intent: declined transfer
'declined transfer' is the correct answer! Good job!

Query: If I'm getting my identity verified, what all do I need?
🦙 Intent: verify top up
The answer 'verify top up' is wrong! You can do better!

Query: Am I allowed to change my PIN anywhere?
🦙 Intent: change pin
'change pin' is the correct answer! Good job!

Query: How do I contact customer support about my declined transfer?
🦙 Intent: contactless not working
```

Figure 1: **Illustration of in-context reinforcement learning**. The context is queries, with predicted model outputs (🦙), and verbalized rewards. The prompt ends with a new example, and the model has to complete its output (red). The model has to learn in-context from the past interactions on the prompt.

---

[1]We use the terms *prompt* and *context* interchangeably.

consisting of input, model output prediction, and the corresponding rewards. As the number of input examples increases, the model observes more triplets, leading to an online learning scenario. These triplets are followed by a new input, for which the model predicts an output. Figure 1 illustrates ICRL prompting. A naive implementation of this formulation is similar to ICL, except that the context grows over time, and instead of relying on annotated labels and a static dataset, the model is exposed to a stream of inputs and rewards.

We study the ICRL capabilities of Llama 3.1 (Llama Team, 2024) and Phi-3.5-mini (Abdin et al., 2024), using several standard classification benchmarks, focusing on single-step RL (i.e., contextual bandits). Unfortunately, as we show, the naive formulation quickly leads LLMs to degenerate, often by always predicting the same output. We identify two causes for this failure. First, LLMs, even when sampled from, exhibit an inability to explore. Second, they struggle to learn from complex in-context signals, such as when rewards are negative.

We address the exploration deficiency by taking advantage of how LLMs are sensitive to their prompt composition in unexpected ways (Chen et al., 2023; Sclar et al., 2024), and add stochasticity to the prompt construction. Given the difficulty of LLMs to benefit from negative examples, we filter these out from the context, making the prompt more stylistically similar to ICL prompts, and easier to reason about. The stochastic prompt effectively neutralizes degeneration in our experiments. Simplifying the prompt further increases performance, even though it removes informative negative learning signals Overall, these mechanisms expose effective ICRL in LLMs.

Our method shows a strong relationship between performance and test-time compute – as compute costs increase, the model also performs better (Snell et al., 2024). Much of the computational cost arises from the stochastic construction of prompts, which occurs for each example and increases in cost as the model observes more examples. To reduce these costs, we show that this process can be approximated, and that the level of approximation is in direct relation to how much compute is allocated to the model.

Overall, we show that our approach is able to overcome the exploration degeneration of both Llama and Phi, leading to impressive and consistent gains through ICRL. For example, in the Banking-77 (Casanueva et al., 2020) classification task, Llama improves from 17.2% zero-shot accuracy to 66.0% through ICRL. We also show that our approximation of ICRL is effective with both models, although the stronger Llama can absorb a much higher approximation level. Our code, data, and experimental logs will be released upon publication.

## 2 IN-CONTEXT REINFORCEMENT LEARNING

ICL operates by providing a model with correct demonstrations of a task. A demonstration includes an input (e.g., *What is the best football club in Europe?*) and its corresponding correct output (e.g., *AC Milan*). In its reliance on gold-standard labels, ICL follows the common supervised learning paradigm, although without any change in the model parameters.

However, in-context learning could also be performed differently. Instead of providing models with correct demonstrations, the model could first try to guess answers, then observe the outcomes (i.e., rewards) of its predictions, and eventually learn from these signals, in an online learning setting, all within the context. This alternative way of learning in context follows the reinforcement learning paradigm (RL; Sutton & Barto, 2018), where models learn by reinforcing good behaviors and suppressing bad choices.

Formally, we are concerned with an RL scenario where the model $\pi$ observes an input $x^{(t)} \sim \mathcal{D}$ sampled from the data distribution $\mathcal{D}$ at time $t$, generates a prediction $\hat{y}^{(t)}$, and then observes a reward $r^{(t)} \sim R(x^{(t)}, \hat{y}^{(t)})$. We denote the tuple $(x^{(t)}, \hat{y}^{(t)}, r^{(t)})$ as an episode.

In common RL terminology, the model $\pi$ is the policy, the input $x^{(t)}$ is the state, and the prediction $y^{(t)}$ is the action. Throughout our formulation, the policy is also conditioned on previous episodes in the form of an LLM context, similar to how supervised examples are provided in ICL. These past episodes are not part of the state. Instead, the context is used to perform in-context policy improvement, similar to how past episodes are used to perform policy improvement in conventional RL (e.g., via parameter updates).

We design several methods to elicit in-context reinforcement learning (ICRL) from LLMs. The naive approach is a straightforward implementation of ICRL following the common ICL recipe (Section 2.1). The explorative approach (Section 2.2) is motivated by the empirical weaknesses we observe in the naive approach (Section 4). Finally, the approximate method comes to reduce some of the computational demands of explorative ICRL.

## 2.1 NAIVE ICRL

Algorithm 1 outlines the most straightforward way to implement ICRL. The model repeatedly observes a new example, predicts its output, and observes its reward. Each such model interaction creates an episode, which is added to an episode buffer. For each interaction, we construct a context $C$ from existing episodes (line 3). As long as the LLM context window allows it, at each time step, all past episodes $\mathcal{E}$ are included in the context in the order they were observed. This allows re-using past computations (i.e., through the KV cache), leading to relatively efficient computation.[2] If the context window length is reached, we only consider recent episodes that fit into the context window, essentially running ICRL with a sliding window as big as the LLM allows.

---

**Algorithm 1** Naive ICRL

**Require:**
  $\mathcal{D}$: Data distribution
  $\pi$: Language model policy
  $R$: Reward function

1: Init buffer $\mathcal{E} \leftarrow \emptyset$
2: **for** $t = 1, 2, 3, \ldots$ **do**
3:   $C \leftarrow$ create context from $\mathcal{E}$
4:   Observe input $x^{(t)} \sim \mathcal{D}$
5:   Sample prediction $\hat{y}^{(t)} \sim \pi(\cdot|C, x^{(t)})$
6:   Observe reward $r^{(t)} \sim R(x^{(t)}, \hat{y}^{(t)})$
7:   Add episode to buffer
      $\mathcal{E} \leftarrow \mathcal{E} \cup \{(x^{(t)}, \hat{y}^{(t)}, r^{(t)})\}$

---

Unfortunately, naive ICRL fails miserably in practice, as we empirically show in Section 4 and Figure 2. Its poor performance is due to its incapacity to explore the output space properly. Figure 3 visualizes how naive ICRL degenerates to predicting just a few labels, far from the real distribution in the data.

## 2.2 EXPLORATIVE ICRL

Explorative ICRL addresses the exploration deficiency observed with naive ICRL by leveraging the sensitivity of LLMs to their prompt. It has been widely observed that changes in prompt composition lead to variance in LLM behavior, including changes in the exact set of examples selected for ICL (Zhang et al., 2022; Liu et al., 2022; Chen et al., 2023; Levy et al., 2023) or even seemingly meaningless stylistic changes (Sclar et al., 2024; Lu et al., 2022).. Generally, this property of LLMs is not considered positively. However, in the case of ICRL, it provides an opportunity to introduce stochasticity into the process, and thereby introducing a level of exploration. We achieve this by randomly choosing the subset of past episodes to include in the prompt each time the model observes a new input.

In addition, we empirically observe that LLMs have a harder time benefiting from negative learning signals (i.e., episodes with negative reward). This has been observed in past feedback-driven continual learning work (Kojima et al., 2021; Suhr & Artzi, 2023). Negative episodes are also not very informative for learning – indicating that one output is bad, essentially encourages an almost uniform distribution over outputs. This leads to the second design decision in explorative ICRL: only include examples with a positive reward in the constructed contexts.

Algorithm 2 describes explorative ICRL. For each input, we construct a new context (lines 3–7). We decide what past episodes to include in this context by sampling from a Bernoulli variable parameterized by $p_{\text{keep}}$ (lines 4–7. We sample independently for each past episode. This results in different reasoning for each input, because each is done with a different context. When storing past episodes, we only include episodes with positive reward (lines 13–14).

---

[2]We assume unbounded memory through our analysis. This makes it possible to compare the computational costs of the different algorithms, because it allows the assumption that it is possible to store the computation for all previous episodes stored in the context.

**Algorithm 2** Explorative ICRL

**Require:**
 $\mathcal{D}$: Data distribution
 $\pi$: Language model policy
 $R$: Reward function
 $p_{\text{keep}}$: Prob. to keep examples in context

1: Init episode buffer $\mathcal{E} \leftarrow \emptyset$
2: **for** $t = 1, 2, 3, \ldots$ **do**
3:  Init empty context $C^{(t)} \leftarrow [\,]$
4:  **for** $e \in \mathcal{E}$ **do**
5:   $b \sim \text{Bernoulli}(p_{\text{keep}})$
6:   **if** $b = 1$ **then**
7:    Add episode to context $C^{(t)} \mathrel{+}= e$
8:  **if** $|C^{(t)}| >$ LLM context window **then**
9:   $C^{(t)} \leftarrow$ downsample $C^{(t)}$
10:  Observe input $x^{(t)} \sim \mathcal{D}$
11:  Sample prediction $\hat{y}^{(t)} \sim \pi(\cdot | C^{(t)}, x^{(t)})$
12:  Observe reward $r^{(t)} \sim R(x^{(t)}, \hat{y}^{(t)})$
13:  **if** $r^{(t)} > 0$ **then**
14:   Add episode to buffer
   $\mathcal{E} \leftarrow \mathcal{E} \cup \{(x^{(t)}, \hat{y^{(t)}}, r^{(t)})\}$

**Algorithm 3** Approximate ICRL

**Require:**
 Everything from Algorithm 2
 $K$: Number of contexts to maintain

1: Init empty contexts $\mathcal{C} \leftarrow \{[\,]^{(1)}, \ldots, [\,]^{(K)}\}$
2: **for** $t = 1, 2, 3, \ldots$ **do**
3:  Sample context uniformly $C \sim \mathcal{U}(\mathcal{C})$
4:  Observe input $x^{(t)} \sim \mathcal{D}$
5:  Sample prediction $\hat{y}^{(t)} \sim \pi(\cdot | C, x^{(t)})$
6:  Observe reward $r^{(t)} \sim R(x^{(t)}, \hat{y}^{(t)})$
7:  **if** $r > 0$ **then**
8:   **for** $k = 1$ to $K$ **do**
9:    $b \sim \text{Bernoulli}(p_{\text{keep}})$
10:    **if** $b = 1$ **then**
11:     Add episode to cached context
     $\mathcal{C}[k] \mathrel{+}= (x^{(t)}, \hat{y}^{(t)}, r^{(t)})$

Depending on $p_{\text{keep}}$, explorative ICRL will encounter the issue of the LLM context window saturating much later than naive. However, deploy ICRL for enough interactions, and the context window will saturate, even for the models with the largest context windows. Similar to naive, we downsample the context if it overflows the LLM context window (line 9). We design three strategies to downsample the context if we reach the limit of the LLM context window: (a) *unbiased*: randomly remove episodes from $C^{(t)}$ until it fits the context window; (b) *start-biased*: use the longest possible prefix of episodes from $C^{(t)}$ such that it fits the LLM context size; and (c) *end-biased*: use the longest possible suffix.

A downside of explorative ICRL is the computational cost. While naive ICRL benefits from caching, caching is not useful for explorative, because the construction of a fresh context $C^{(t)}$ for each episode eliminates this option for explorative ICRL. The probability of encountering the same context twice, or even the same prefix, is exceptionally low even after a few episodes. This means that the context has to be computed from scratch for each input. If $p_{\text{keep}}$ is relatively small, contexts are likely to be much shorter than in naive. Although this can reduce the cost, explorative remains much more computationally demanding than naive.

## 2.3 APPROXIMATE ICRL

Explorative ICRL addresses the exploration deficiencies of naive ICRL, but incurs high computational costs (Section 2.2). We propose an approximation of explorative ICRL that strikes a balance between computational cost and learning effectiveness. Similar to explorative ICRL, the approximate version also excludes episodes with negative reward and focuses on exploration by stochasticity in the context.

Algorithm 3 describes approximate ICRL. The core idea behind the approximation is to limit the number of contexts, so we can simply gradually expand them with new episodes, rather than always create and compute new contexts. We maintain $K$ contexts $\mathcal{C}$, which all start empty (line 1). At each time step $t$, we sample a context $C$ from the $K$ contexts (line 3), and use it for episode $t$ (lines 4–6. If the reward $r^{(t)} > 0$, we use the episode to expand all contexts stochastically. For each context in $\mathcal{C}$, we expand it with the $t$-th episode with a probability of $p_{\text{keep}}$ (lines 8–11).

Approximate ICRL introduces stochasticity in two places: sampling the context to use for each episode and the expansion of the stored contexts. In Algorithm 3, we use *uniform* sampling to choose the context (line 3). This is a uniform approximation of the probability of a context, which can also be easily computed *exactly* using the probabilities of the episodes it contains and $p_{\text{keep}}$. In practice,

we find the exact computation to work poorly, because contexts that are assigned more episodes or have low probability episodes quickly receive very low probability, and are not used. Figure 5b shows this experimental analysis. We use uniform sampling throughout our experiments.

The level of approximation the algorithm provides depends on the resources available. For example, one can allocate each context to a compute unit, so a machine with eight compute units (e.g., GPUs) will support $K = 8$. Approximate ICRL is a strict approximation of explorative ICRL in the sense that coupling the exact context sampling strategy and $K \to \infty$ gives explorative ICRL. However, the approximation is limited in handling contexts that extend beyond the LLM window length. Overcoming this while maintaining the efficiency of the approximation is an important direction for future work.

## 3 EXPERIMENTAL SETUP

**Models**  We use the instruction-tuned versions of Llama 3.1 8B (Llama Team, 2024) and Phi-3.5-mini 3.8B (Abdin et al., 2024). We chose these model families because, at the time of this work, they are the only popular open-source large language models that support more than 100k tokens in the context, while still having less than 10B parameters. We use both in a chat-like format, with multiple turns. We compute the maximum number of episodes the context window can take for each model and task combination. Appendix A.2 reports the exact numbers. Both models are used for the main experiments, but only Llama for a few secondary experiments, due to the computational costs. We use constrained decoding when generating model predictions, similar to recent work on ICL (Bertsch et al., 2024).

**Tasks**  We follow Bertsch et al.'s (2024) study of many-shot ICL in focusing on five classification problems: Banking-77 (77 labels; Casanueva et al., 2020), Clinic-150 (150 labels; Larson et al., 2019), NLU (68 labels; Liu et al., 2021), TREC (6 labels: Li & Roth, 2002; Hovy et al., 2001), and TREC-fine (50 labels; Li & Roth, 2002; Hovy et al., 2001). Because of the large output spaces (up to 150 labels in Clinic-150), these tasks are challenging for large language models, as empirically shown by Bertsch et al. (2024) and replicated in our ICL experiments.

The datasets are of different sizes. The size of the datasets dictates the number of time steps in our experiments. We randomly sub-sample Banking-77, Clinic-150, and NLU to 10k examples. TREC and TREC-fine are smaller, so we only use 5k training examples for each. This allows the experiments to be of relatively standard length. The training data corresponds to the data distribution $\mathcal{D}$ in our algorithms. We also sub-sample all test sets to 500 examples each, to reduce the computational cost of experiments. NLU does not provide a standard test set, so we create our own train and test splits. In all experiments, the datasets contain the same examples in the same order.

**Rewards and Prompt Design**  We use a deterministic binary reward function. Rewards are computed from the gold-standard labels in the dataset. We automatically transform the numerical rewards into a natural language format indicating if the model prediction is correct or not, which is more suitable for LLM reasoning. Appendix A.1 provides more details about our prompting decisions.

**Evaluation**  We report running test accuracy. For test accuracy, we use the held-out test set of each dataset. We compute it every 500 steps for each test example separately, using the context used to process that step's training example. In some cases, we also report train accuracy as the running mean accuracy over the most recent 256 episodes.

We also report regret, the forgone utility from an actual model prediction in comparison to the oracle choice. Intuitively, regret measures how many interactions the model handled poorly throughout the experiment. In our experiments, regret is the accumulated number of incorrect examples throughout learning. Regret gives a single number that considers both the final performance and how fast the model reached it. A good system would reach high performance as fast as possible, making fewer mistakes overall (i.e., would have a low regret).

**Comparisons**  We compare our ICRL algorithms against the zero-shot setting, which corresponds to the performance on the test set at step zero (i.e., without any in-context examples). We also report supervised ICL performance for all tasks to contextualize the results. We generally expect supervised

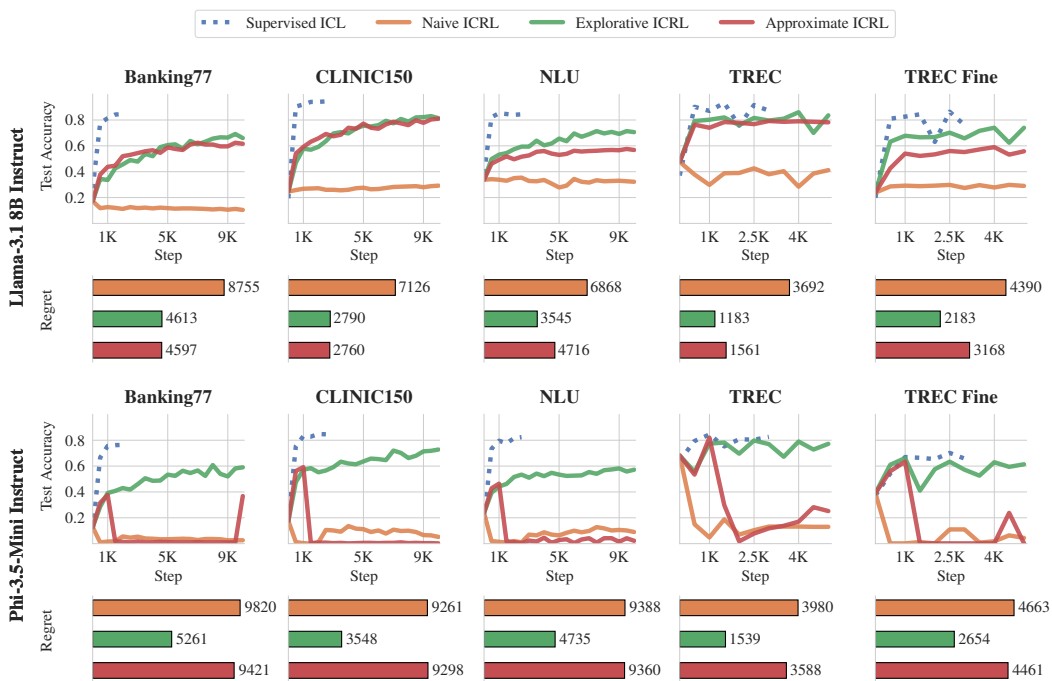

Figure 2: **Performance of ICRL**. Naive, Explorative, and Approximate held-out test results and regret for both models and all tasks. We also report Supervised ICL performance. Explorative consistently outperforms zero-shot (i.e., first step) and Naive, while also showing consistent trends of continual improvement as more data is observed.

ICL to outperform ICRL, because it has access to gold-standard labels. In particular, at each time step where we report supervised ICL performance, we provide the model with all examples observed so far by the ICRL methods, but with gold-standard labels. We stop the supervised ICL experiments when the number of examples becomes bigger than the maximum number supported by the context window.

## 4 RESULTS AND ANALYSIS

We show the test accuracies and training regrets Figure 2.[3] We also show the performance of supervised ICL for comparison, although it relies on a much higher degree of supervision.

**LLMs Can Learn In-Context From Rewards Alone** Explorative effectively learns in all tasks and for both models, showing significant improvements over zero-shot. Explorative improves over the performance of zero-shot Llama by +48.8% in Banking-77, +56.8% in Clinic-150, +36.8% in NLU, +36.0% in TREC, and +50.2% in TREC-fine; and the same with Phi by +46.2% in Banking-77, +55.2% in Clinic-150, +33.4% in NLU, +9% in TREC, and +22.4% in TREC-fine. In general, its accuracy approaches the supervised ICL upper bound in some settings, and it always outperforms zero-shot. For both models, Explorative also demonstrates a continual growth in performance over time, suggesting that with more data its performance would improve. This is especially evident for the most challenging datasets, i.e., the ones with the most labels (i.e., Banking-77, Clinic-150, NLU), as they require a much stronger exploration effort. Thus, our empirical findings show that LLMs can learn in-context from rewards alone.

**Naive Fails to Explore** The Naive does not learn and in most cases even deteriorates below zero-shot (Figure 2). One key issue is exploration. Figure 3 shows prediction confusion matrices, output distributions, and data distributions for the Banking-77 task with Llama, comparing zero-short, Naive,

---

[3]Unless specified otherwise, we use: $p_{keep} = 0.1$; uniform context sampling and $K = 8$ for Approximate; unbiased downsampling when the context fills for Explorative.

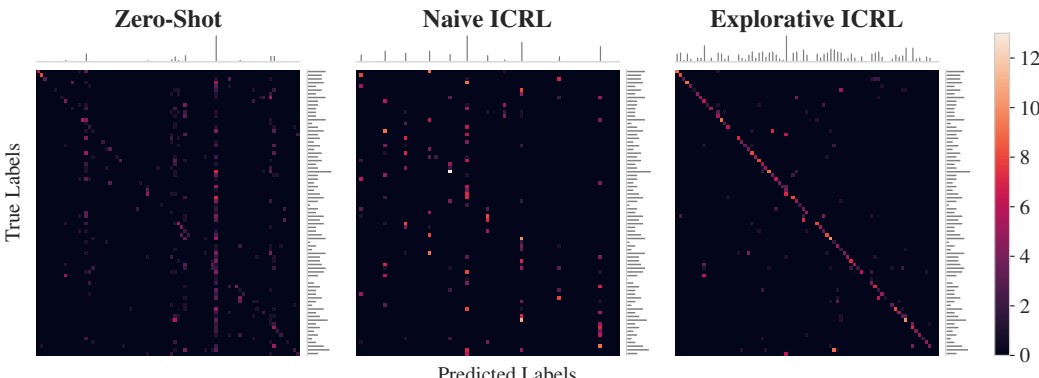

Figure 3: **Confusion matrices for zero-shot, Naive and Explorative**. Each position $(x, y)$ represents the number of times Llama predicts label $x$ while the true label is $y$ in Banking-77 on the test data. For Naive and Explorative, we report results at the final time step. On the right of each matrix, we report the distribution of true labels. On top of each matrix, we report the distribution of predicted labels. The distributions are not on the same scale for visibility. Only Explorative does not present a skewed distribution and is concentrated along the diagonal, meaning most times the predicted label is the correct label.

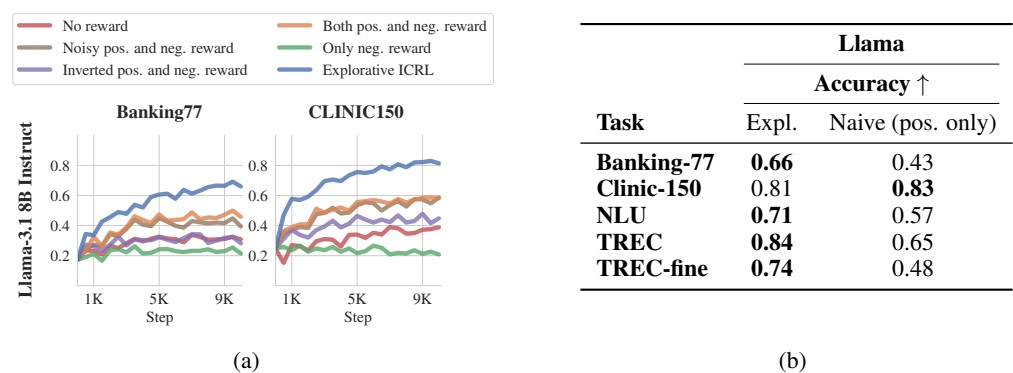

| Legend | | |
|---|---|---|
| No reward | Both pos. and neg. reward | |
| Noisy pos. and neg. reward | Only neg. reward | |
| Inverted pos. and neg. reward | Explorative ICRL | |

| | Llama | |
|---|---|---|
| | **Accuracy ↑** | |
| **Task** | Expl. | Naive (pos. only) |
| **Banking-77** | **0.66** | 0.43 |
| **Clinic-150** | 0.81 | **0.83** |
| **NLU** | **0.71** | 0.57 |
| **TREC** | **0.84** | 0.65 |
| **TREC-fine** | **0.74** | 0.48 |

(a)                                             (b)

Figure 4: **Explorative ICRL ablations**. (a) Test accuracy of Explorative ICRL with different reward signals. Positive reward only is the best choice. (b) Comparison of Explorative with Naive when only positive rewards are used (ablating on the stochastic context from Explorative). Test accuracy is reported at the final step. Explorative consistently outperforms Naive with positive rewards only by large margins, except for one case where they are tied.

and Explorative. A perfect classifier would have non-zero counts only on the diagonal, and the output distribution would be identical to the data distribution. Both Naive and Explorative start from zero-shot. After learning, Explorative shows a clear focus on the diagonal and higher similarity between the prediction and data distributions. Naive fails to learn to effectively classify. The output distribution explains why: its focus on just a few labels indicates it failed to explore.

**Both Modifications of Explorative are Important**  Explorative modifies Naive in two ways: stochasticity for exploration and episodes with positive rewards to simplify the context. Explorative with both positive and negative rewards learns, but much less effectively than if we omit episodes with negative rewards (Figure 4a). On the other hand, Figure 4b shows that even though omitting negative examples from Naive helps, there remains a large gap to Explorative. Figure 4a also shows the impact of reward. We see some level of learning without rewards or with inverted rewards. This aligns with past observations of a domain effect in ICL (Min et al., 2022; Pan et al., 2023a; Lyu et al., 2023; Kossen et al., 2024). However, this learning is relatively minimal, and including both positive and negative episodes improves performance significantly, albeit it remains much lower than with positive episodes only. Including only negative episodes, on the other hand, is catastrophic.

We also observe that when providing noisy rewards (i.e., with a probability of 10% the reward is inverted) performance does not degrade significantly, suggesting that ICRL has some robustness to environments with noisy learning signals. Of course, this is an initial experiment of robustness to noise, and we leave a more detailed analysis for future work.

**Uniform Context Sampling in Approximate is Better**    We observe empirically that exact context sampling in Approximate performs worse than uniform sampling. This happens because exact computation likely leads to always using the same context at later steps, as small changes in the probability of sampling can compound once the model is biased towards one context. Table 5b reports regret and final accuracy for exact and uniform strategies, on Llama and both Banking-77 and Clinic-150 tasks. Appendix B provides more details on this comparison, including visualization of context selection.

**Approximate is an Effective Alternative to Explorative**    In the Figure 2, Approximate performs almost as well as Explorative ICRL when trained with Llama, across all tasks. The results are very different with Phi: despite early learning, Approximate deteriorates quickly. This stems from one of the contexts being biased toward one label and therefore predicting only this label. Eventually, episodes with this label spread to other contexts, leading to the collapse in performance we observe. It is empirically possible to recover, as we see in Banking-77 later in the experiment, but the chance of it happening seems very low. The success of Llama and failure of Phi with $K = 8$ show that different LLMs have different sensitivity to the approximation. Figure 5a shows that that with a higher number of contexts $K > 32$ Phi is able to effectively learn, indicating Phi needs a higher computational budget. Figure 13 in Appendix B shows this sensitivity analysis for Llama. Overall, Llama is robust to the approximation, with most values performing similarly to Explorative, except with the lowest values of $K$.

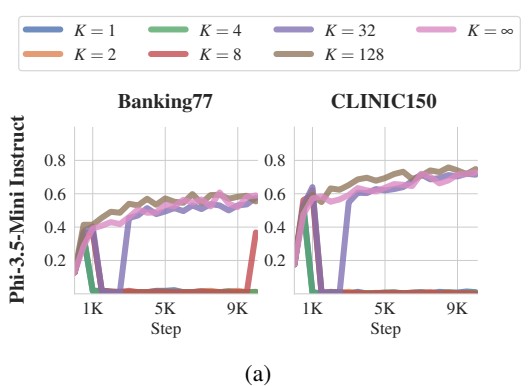

(a)

|  | Accuracy ↑ | | Regret ↓ | |
|---|---|---|---|---|
| **Task** | Exact | Unif. | Exact | Unif. |
| **Banking-77** | 0.55 | **0.62** | 5108 | **4597** |
| **Clinic-150** | 0.73 | **0.81** | 3301 | **2760** |

(b)

Figure 5: **Comparison of Approximate parameters**. (a) Effect of the number of contexts $K$. We report test accuracy for Phi only, as it proves more sensitive to this approximation. Generation degenerates for low $K$, while the model can learn for $K \geq 32$. (b) Comparison of exact and uniform sampling. We report test accuracy at the final step and regret for Llama. Uniform sampling strategy is consistently better.

**Approximate Reduces Compute Needs**    We measure the reduction of tokens processed in Approximate compared to Explorative throughout full ICRL runs. We approximate this measure by computing at each step the number of tokens required for a forward call and subtracting the number of tokens of the sequence with the longest common prefix processed in a previous step, as it would be possible to use the KV cache for all the tokens in the common prefix (assuming infinite memory). We find that Explorative processes two orders of magnitude more tokens than Approximate. Table 2 in Appendix B provides numerical results for this analysis.

**ICRL is Sensitive to Stochasticity Level**    Stochasticity in context generation is one of the important components that contribute to both Explorative and Approximate performance. It is modulated by setting $p_{\text{keep}}$. Figure 6 shows the sensitivity of Explorative to the value of $p_{\text{keep}}$. Without stochasticity ($p_{\text{keep}} = 1.0$), ICRL struggles on both models, but especially on Phi. However, if $p_{\text{keep}}$ is too high, we retain too few examples on the context, and it can hurt performance.

**Comparison of Context Subsampling Strategies**    In practice, we never saturate the LLM context window when using Llama or Phi because our context window is more than 100k and $p_{\text{keep}} = 0.1$. We conduct experiments to evaluate the strategies we presented in Section 2.2 to handle the case of overflowing the context window by limiting the context window of Llama to 4k or 8k tokens.

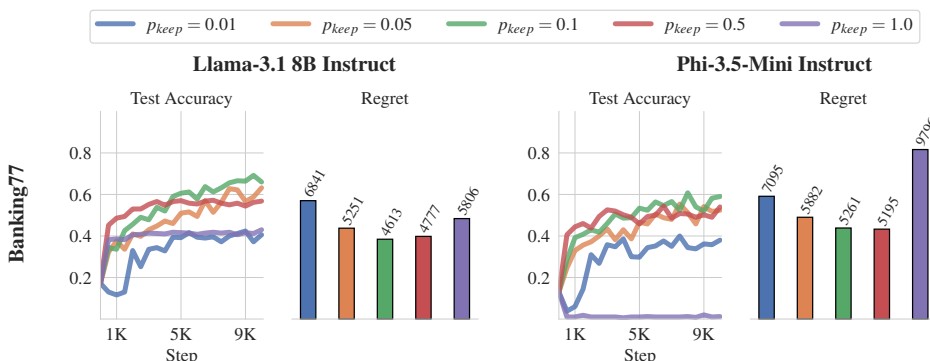

Figure 6: **Sensitivity to $p_{keep}$ in Explorative ICRL**. We compare performance with different values of $p_{keep}$. Intermediate values learn better for both models.

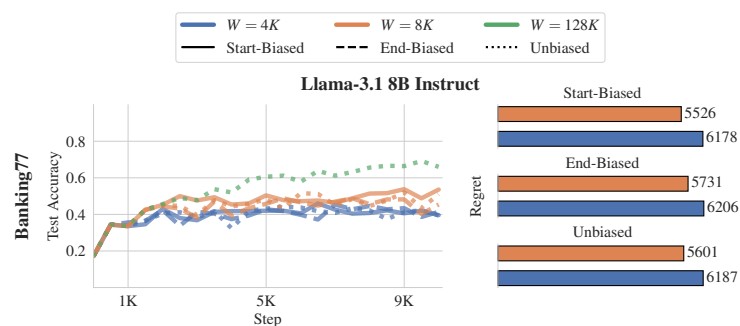

Figure 7: **Performance with limited context and effect of subsampling strategies**. We report test accuracy and regret of Llama with Banking-77. As expected, a longer context leads to better performance. The difference between the sampling strategies is negligible, with start-biased slightly better.

Generally, we observe that *start-biased* strategy outperforms *unbiased*, which in turn performs better than *end-biased*, in all cases, although by only small margins. Figure 7 shows the results of this analysis for Banking-77, and Figure 12 in Appendix B for Clinic-150.

## 5 RELATED WORK

**In-Context (Supervised) Learning**   ICL was first demonstrated by Brown et al. (2020), and since then its causes (Chan et al., 2022; Xie et al., 2022; Olsson et al., 2022; Garg et al., 2022; Von Oswald et al., 2023; Hendel et al., 2023; Wang et al., 2023) and the level of learning it displays (Min et al., 2022; Lyu et al., 2023) have been studied extensively. By now, it is well established that LLMs can learn new tasks in context (Garg et al., 2022; Wei et al., 2023; Pan et al., 2023b; Kossen et al., 2024; Li et al., 2024). Our work builds on this line of work, and provides the first evidence that LLMs can perform RL in context, and not only supervised learning (i.e., the standard way it is done).

Our study would not be possible without recent increases in the context window length of LLMs (Llama Team, 2024; Abdin et al., 2024; Gemini Team, 2024). Recent work showed that model performance can continue to increase when including hundreds or thousands of demonstrations (Bertsch et al., 2024; Agarwal et al., 2024). We find similar results, as LLMs can continually improve when learning through ICRL until their context does not saturate. Interestingly, while some work (Zhang et al., 2024; Mo et al., 2024; Shinn et al., 2024) find that models can learn from mistakes, our results do not support this. It is possible that models can learn from mistakes only when explicitly reasoning (Kojima et al., 2022; Wei et al., 2022) about them (Zhang et al., 2024; Shinn et al., 2024) and cannot implicitly leverage negative signals.

**In-Context Reinforcement Learning** Likely the closest work to ours is Krishnamurthy et al. (2024). They investigate whether LLMs can solve multi-armed bandit problems, a state-less simpler RL setting than the one we are focused on. Our experimental setup revolves around contextual bandit problems, where the best action depends on the specific input. We observe similar issues to their findings with the Naive approach. They present a set of negative results, and finally are able to elicit effective learning, but through a prompting strategy that cannot generalize beyond their very simple scenario. They conclude LLMs cannot explore, similar to our conclusions from Naive. However, we address this problem by developing Explorative, which includes stochasticity and focuses on positive episodes. Wu et al. (2024) propose a set of benchmarks that includes a simplified multi-armed bandit problem. The provide baseline performance with a method similar to Naive, showing mixed results even given the extreme simplicity of their setting, compared to that of Krishnamurthy et al. (2024).

Another related line of research is that of Transformers trained to solve sequential decision-making problems (Janner et al., 2021; Chen et al., 2021; Xu et al., 2022; Laskin et al., 2022; Zheng et al., 2022; Lee et al., 2023; Grigsby et al., 2023; Raparthy et al., 2023). In all these cases, Transformers (Vaswani, 2017) are trained from scratch. Our focus is different: we study ICRL that emerges from the process of training LLMs, without fine-tuning the LLM for this purpose.

# 6 DISCUSSION AND LIMITATIONS

We study the potential of LLMs to demonstrate ICRL, and propose several algorithms to elicit this kind of behavior: Naive, Explorative, and Approximate. Naive fails miserably, but this allows us to identify exploration as the key missing ingredient. Explorative introduces stochasticity to the prompt construction, and combined with focusing on positive examples, shows consistent ICRL. However, this comes at a high computational cost. The third algorithm we proposes, Approximate, comes to address this cost, by a strict approximation of Explorative. We provide a detailed analysis of the various methods, and show the importance of each of our choices, and the sensitivity of the process to various settings.

Our work carries several limitations, all outline important directions for future work. The first is due to our choice of problems to study. We intentionally selected classification benchmarks to simplify the experiments and evaluation in this early stage of studying ICRL. However, this leaves open the question of applicability to more complex problems, where rewards are more nuanced. For example, summarization and question answering provide much more formidable challenges, albeit with complex evaluation challenges. We believe our work enables future work to study these challenges, and that this is an important direction.

Another limitation, is our use of a binary reward function. This choice directly falls off our choice of classification problems. It is another important aspect in making our benchmark environment straightforward to experiment with. However, it leaves a very important question open: can ICRL handle more nuanced reward signals? For example, a reward function that can give all possible real number in a specific range. Such a reward function leads to an interesting challenge in decoding it into language. It is a particularly important question given our findings with regard to the ability of LLMs to learn in-context from negative rewards. The problem we identified with reasoning about episodes with negative rewards pose another limitation, and lays out an important research question for future work.

Our work also lays out open questions as far as the use of computational resources. Our methods are relatively compute intensive, especially after the learner observes many episodes. We propose Approximate to address this, and show how it allows to trade-off compute for robustness. However, Approximate left open the problem of working with limited context window, a critical problem for deploying these methods for extended periods with many interactions. This, again, is a very important direction for future work.

Finally, not a limitation per se, but we kept prompt optimization to a minimum. This was an intentional choice, because our goal is to find robust behaviors, and not prompt engineer the problem. However, this does leave significant room for development, likely improving on the results we observe.

We hope our work helps to shed light on the capabilities of contemporary LLMs, and that it lays our the ground for extensive future work, both research and practice.

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

## A EXPERIMENTAL SETUP

Each experiment is conducted on a node equipped with four NVIDIA A100 GPUs, each with 40GB of memory. For efficient inference, we employ the *vllm* library (Kwon et al., 2023).

### A.1 PROMPT DESIGN

We report prompt examples from ICL and ICRL experiments. We show the prompts for both Llama and Phi, because *transformers* library (Wolf et al., 2019), which we use for the tokenizers, automatically injects the cut-off and current dates in Llama's system prompt, making it slightly different from that of Phi. In all cases, we show the prompts with two in-context examples.

---

**Prompt example for ICL in Llama**

```
<|begin_of_text|><|start_header_id|>system<|end_header_id|>\n\nC
↪  utting Knowledge Date: December 2023\nToday Date: 26 Jul
↪  2024\n\nYou are an useful assistant. Answer the following
↪  questions.<|eot_id|><|start_header_id|>user<|end_header_id|>
↪  \n\nQuery: Tell me about the card
↪  PIN?<|eot_id|><|start_header_id|>assistant<|end_header_id|>\
↪  n\nIntent: get physical
↪  card<|eot_id|><|start_header_id|>user<|end_header_id|>\n\nQu
↪  ery: Is there a daily auto top-up
↪  limit?<|eot_id|><|start_header_id|>assistant<|end_header_id|
↪  >\n\nIntent: automatic top
↪  up<|eot_id|><|start_header_id|>user<|end_header_id|>\n\nQuer
↪  y: I got a message saying I made a withdrawal from the bank
↪  machine, but I did not.<|eot_id|><|start_header_id|>assistan
↪  t<|end_header_id|>\n\nIntent:
```

Figure 8: **An example of prompt of ICL for Llama.**

---

**Prompt example for ICRL in Llama**

```
<|begin_of_text|><|start_header_id|>system<|end_header_id|>\n\nC
↪  utting Knowledge Date: December 2023\nToday Date: 26 Jul
↪  2024\n\nYou are an useful assistant. Answer the following
↪  questions. Feedback will indicate if you answered correctly.
↪  You must answer correctly, using previous feedback to make
↪  better predictions.<|eot_id|><|start_header_id|>user<|end_he
↪  ader_id|>\n\nQuery: what's the traffic at
↪  lexington<|eot_id|><|start_header_id|>assistant<|end_header_
↪  id|>\n\nIntent:
↪  traffic<|eot_id|><|start_header_id|>user<|end_header_id|>\n\
↪  n'traffic' is the correct answer! Good job!\n\nQuery: my
↪  credit card is set to expire in what month<|eot_id|><|start_
↪  header_id|>assistant<|end_header_id|>\n\nIntent: expiration
↪  date<|eot_id|><|start_header_id|>user<|end_header_id|>\n\n'e
↪  xpiration date' is the correct answer! Good job!\n\nQuery:
↪  could you translate atm machine into english<|eot_id|><|star
↪  t_header_id|>assistant<|end_header_id|>\n\nIntent:
```

Figure 9: **An example of prompt of ICRL for Llama.**

---

**Prompt example for ICL in Phi**

```
<|system|>\nYou are an useful assistant. Answer the following
↪    questions.\n<|end|>\n<|user|>\nQuery: what's the traffic at
↪    lexington<|end|>\n<|assistant|>\nIntent:
↪    traffic<|end|>\n<|user|>\nQuery: what is 8
↪    factorial<|end|>\n<|assistant|>\nIntent:
↪    calculator<|end|>\n<|user|>\nQuery: correct, that's
↪    true<|end|>\n<|assistant|>\nIntent:
```

Figure 10: **An example of prompt of ICL for Phi.**

---

**Prompt example for ICRL in Phi**

```
<|system|>\nYou are an useful assistant. Answer the following
↪    questions. Feedback will indicate if you answered correctly.
↪    You must answer correctly, using previous feedback to make
↪    better predictions.\n<|end|>\n<|user|>\nUtterance: meeting
↪    next week monday<|end|>\n<|assistant|>\nIntent: calendar
↪    query<|end|>\n<|user|>\nThe answer 'calendar query' is wrong!
↪    You can do better!\n\nUtterance: how warm today
↪    is<|end|>\n<|assistant|>\nIntent: weather
↪    query<|end|>\n<|user|>\n'weather query' is the correct
↪    answer! Good job!\n\nUtterance: hey make sure i go to sarahs
↪    birthday party on the twelveth<|end|>\n<|assistant|>\nIntent:
```

Figure 11: **An example of prompt of ICRL for Phi.**

## A.2 CONTEXT WINDOWS

For each task and model combination, we conservatively estimate the maximum number of examples that could fit within the context window. This is done by including all observed examples in descending order of token count in the prompt, assuming the model consistently responds with the longest label and that the formatted reward message is at its maximum length. We perform this calculation using the maximum context window for both Llama and Phi. Additionally, for Llama, we repeat the process with context windows of 4096 and 8192 tokens specifically for the Banking-77 and Clinic-150 tasks. All results are reported in Table 1.

| | Phi | Llama | | |
|---|---|---|---|---|
| **Task** | 128k tokens | 4k tokens | 8k tokens | 128k tokens |
| **Banking-77** | 1538 | 34 | 74 | 1673 |
| **Clinic-150** | 2241 | 60 | 126 | 2384 |
| **NLU** | 2397 | - | - | 2425 |
| **TREC** | 2848 | - | - | 2919 |
| **TREC-fine** | 2584 | - | - | 2776 |

Table 1: **Number of maximum examples supported by model and task, given a specific context window**. We compute the numbers of maximum examples supported by a context window of 128k tokens for both Llama and Phi and all tasks, but also of 4k and 8k tokens for Llama, with Banking-77 and Clinic-150 only.

## B ADDITIONAL RESULTS

| Task | Phi | | | Llama | | |
|---|---|---|---|---|---|---|
| | Expl. | Approx. | Ratio | Expl. | Approx. | Ratio |
| **Banking-77** | 87,369,607 | 510,786 | 171 | 102,282,989 | 539,367 | 190 |
| **Clinic-150** | 105,545,002 | 398,677 | 265 | 122,455,599 | 440,019 | 278 |
| **NLU** | 89,894,548 | 409,680 | 219 | 114,517,653 | 433,254 | 264 |
| **TREC** | 29,306,971 | 212,855 | 138 | 34,509,170 | 229,046 | 151 |
| **TREC-fine** | 20,658,980 | 222,955 | 93 | 25,522,358 | 234,884 | 109 |

Table 2: **Reduction of tokens processed in Approximate compared to Explorative throughout full ICRL runs**. Explorative processes two orders of magnitude more tokens than Approximate.

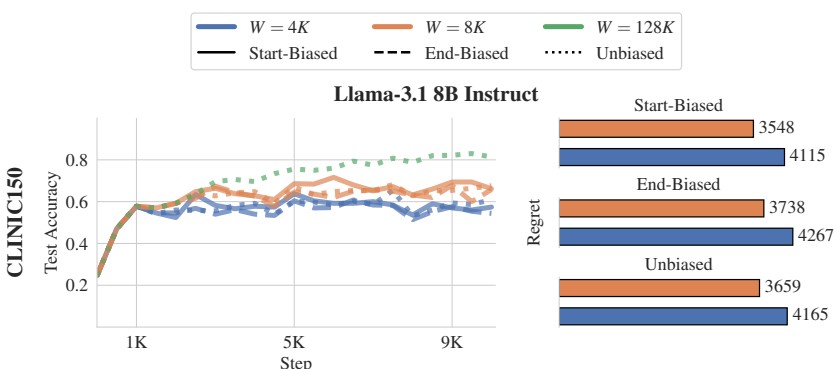

Figure 12: **Performance on Limited Context and Effect of Subsampling Strategies**. We report test accuracy and regret of Llama with Clinic-150. As expected, longer context leads to better results, while early examples seem more important.

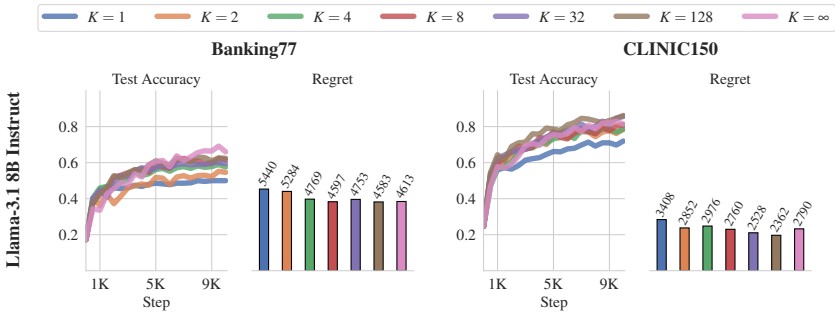

Figure 13: **Effect of the number of contexts $K$ in Approximate on Llama**. The model can learn with all $K$, although higher values perform better.

| | Llama | | | | Phi | | | |
|---|---|---|---|---|---|---|---|---|
| | Accuracy ↑ | | Regret ↓ | | Accuracy ↑ | | Regret ↓ | |
| Task | Expl. | Naive (pos. only) | Expl. | Naive (pos. only) | Expl. | Naive (pos. only) | Expl. | Naive (pos. only) |
| **Banking-77** | **0.66** | 0.43 | **4613** | 5800 | **0.59** | 0.01 | **5261** | 9796 |
| **Clinic-150** | 0.81 | **0.83** | 2790 | **2364** | **0.73** | 0.00 | **3548** | 9824 |
| **NLU** | **0.71** | 0.57 | **3545** | 4608 | **0.57** | 0.02 | **4735** | 9738 |
| **TREC** | **0.84** | 0.65 | **1183** | 2508 | **0.77** | 0.28 | **1539** | 4045 |
| **TREC-fine** | **0.74** | 0.48 | **2183** | 3470 | **0.61** | 0.01 | **2654** | 4798 |

Table 3: **Comparison of Explorative with Naive when only positive rewards are used** (ablating on the stochastic context from Explorative). Test accuracy is reported at the final step. Explorative consistently outperforms Naive with positive rewards only by large margins, except for one case where they are tied.

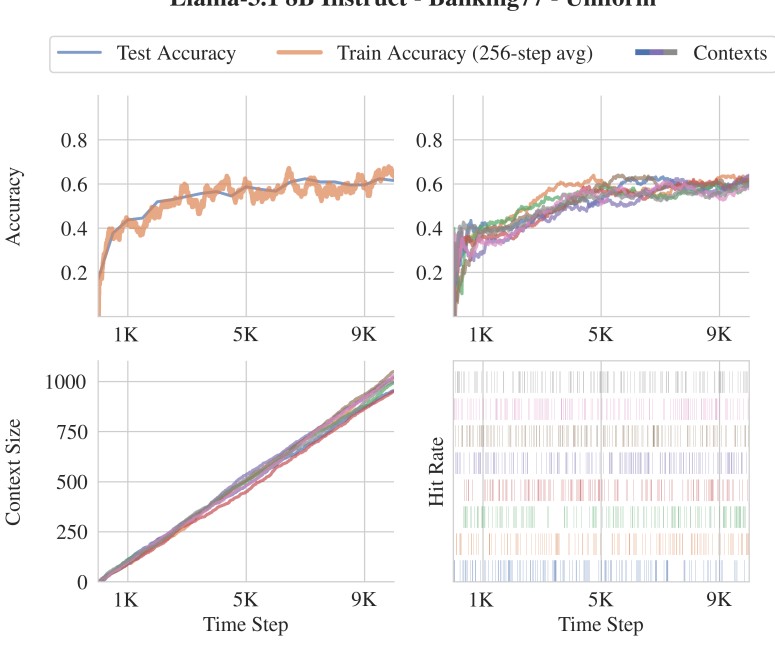

Figure 14: **Detailed visualization of Approximate for Llama, Banking-77 with exact context sampling.** We report test accuracy (top left), a 256-step running average of the training accuracy (bottom left), the training accuracy of each context (top right), and the hit rate of each context (bottom right).

Figure 15: **Detailed visualization of Approximate for Llama, Banking-77 with uniform context sampling.** We report test accuracy (top left), a 256-step running average of the training accuracy (bottom left), the training accuracy of each context (top right), and the hit rate of each context (bottom right).

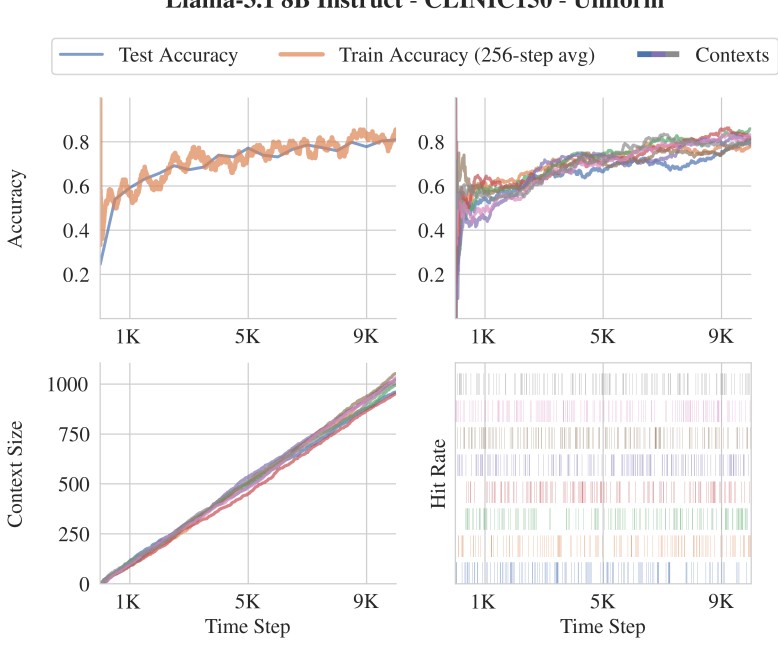

Figure 16: **Detailed visualization of Approximate for Llama, Clinic-150 with exact context sampling.** We report test accuracy (top left), a 256-step running average of the training accuracy (bottom left), the training accuracy of each context (top right), and the hit rate of each context (bottom right).

Figure 17: **Detailed visualization of Approximate for Llama, Clinic-150 with uniform context sampling.** We report test accuracy (top left), a 256-step running average of the training accuracy (bottom left), the training accuracy of each context (top right), and the hit rate of each context (bottom right).

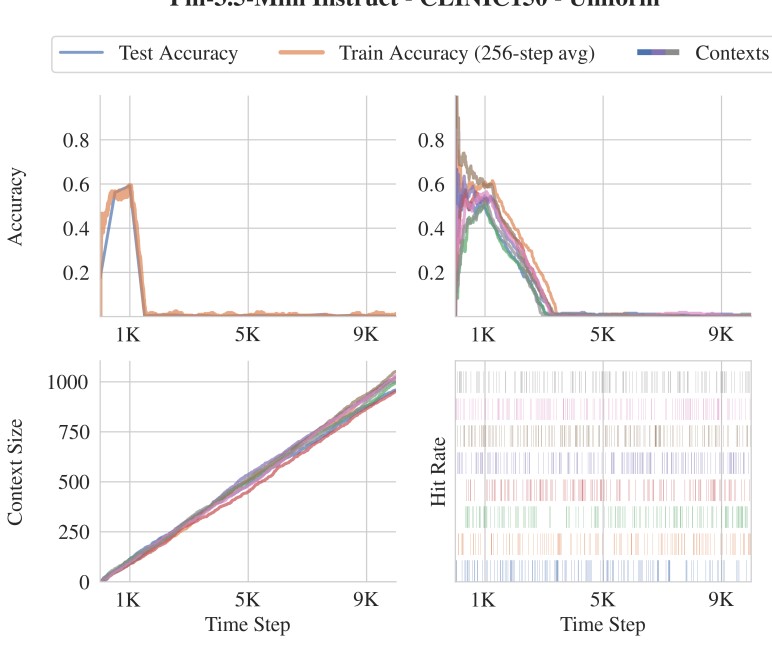

Figure 18: **Detailed visualization of Approximate for Phi, Banking-77 with uniform context sampling.** We report test accuracy (top left), a 256-step running average of the training accuracy (bottom left), the training accuracy of each context (top right), and the hit rate of each context (bottom right).

Figure 19: **Detailed visualization of Approximate for Phi, Clinic-150 with uniform context sampling.** We report test accuracy (top left), a 256-step running average of the training accuracy (bottom left), the training accuracy of each context (top right), and the hit rate of each context (bottom right).

