# OpenReview forum: "LLMs Are In-Context Reinforcement Learners"
_ICLR.cc/2025/Conference — ICLR 2025 Conference Withdrawn Submission_

### Official Review · Reviewer_bCcG · 2024-11-01

**Soundness:** 3
**Presentation:** 3
**Contribution:** 1
**Rating:** 3
**Confidence:** 4

**Summary:**

This paper investigates the in-context reinforcement learning (ICRL) capability of large language models (LLMs). First, a naive ICRL approach was studied, where training pairs and rewards were added to the LLM's context after each step. This naive implementation showed very poor performance on the benchmark tasks. The authors then proposed two sampling-based variants referred to as Explorative ICRL and Approximate ICRL. These methods involve randomly removing training pairs and rewards from the LLM's context, motivated by the idea that incorporating randomness should improve exploration. Both variants demonstrated significantly better results on benchmark tasks compared to the naive approach, although they did not reach the level of supervised training. Approximate ICRL is a more efficient form of Explorative ICRL, with only a minor effect on performance. Ablation studies examined the types of rewards retained in context, and showed that keeping only positive rewards was the best. Other ablations on the keep probability in the proposed variant and context subsampling strategies are also reported.

**Strengths:**

This paper demonstrates significant performance improvements over naive ICRL on the benchmarked tasks. The experimental results show that both Explorative ICRL and Approximate ICRL outperform the naive implementation.

The paper also provides experimental evidence to show that under-exploration is a key reason for poor performance in the naive implementation of ICRL. Moreover, that reward selection also affects ICRL performance and that keeping only positive rewards lead to better performance. Both of these insights are interesting and well supported.

The proposed methods, Explorative ICRL and Approximate ICRL, are novel and effective. These techniques introduce randomness into the context to encourage exploration, while being simple to implement. Approximate ICRL is also not very computationally expensive.

One key strength of this work is the extensive set of benchmark experiments and ablation studies, covering types of rewards, sampling probabilities, and context subsampling strategies. The contribution and influence of each element in the proposed methods are well understood and characterized.

**Weaknesses:**

The motivation for ICRL is unclear for the settings described in the paper. As shown in Figure 2, all of the tasks studied in the paper performed better with supervised learning (SL). Since the reward is based on ground-truth labels, the setting studied requires access to ground-truth, hence is equivalent to SL. Since ICRL is not shown to exceed in-context SL, it is difficult to understand why ICRL should be used in this setting over SL. Are there settings where it makes sense to use ICRL over SL?

The authors found that filtering for examples with only positive rewards improves ICRL. Again, since the reward is based on ground-truth labels, this could just be reconstructing the SL training set in context, making the results trivial. Moreover, STaR[0]-like methods that sample->filter by reward->add to context is known to work quite well in-context [1], reducing the novelty of this work.

This work only focuses on single-step RL and not multi-step sequential decision making problems. This limitation should be made clear. The paper should be careful to not over-claim about the in-context RL capabilities that it has shown vs the more classical multi-turn RL setting, which it does not study. Perhaps "Contextual Bandits" should be included in the title rather than "Reinforcement Learner".

Finally, there are existing works on inference-only in-context RL that study multi-step sequential decision making problems [2, 3]. This highlights that the scope of this paper is unnecessarily restrictive and weakens the novelty of its contributions. The authors should cast the contributions of this paper relative to these prior works.

[0] STaR: Bootstrapping Reasoning With Reasoning, Zelikman et al.
[1] Many-shot In-Context Learning, Agarwal et al.
[2] Large Language Models can Implement Policy Iteration, Brooks et al.
[3] Large Language Models as General Pattern Machines, Mirchandani et al.

**Questions:**

Why should one use ICRL rather than SL in these settings?

Can you provide more details on the benchmark tasks used? In the appendix and not only as a reference?

In Figure 2, is the zero-shot comparison just the starting point (1st step) in the plots? Is this a fair comparison? Shouldn’t the in-context SL be the comparison baseline instead?

---

### Official Review · Reviewer_oPej · 2024-11-03

**Soundness:** 2
**Presentation:** 3
**Contribution:** 2
**Rating:** 3
**Confidence:** 4

**Summary:**

This paper demonstrates the use of LLMs to work with in-context reinforcement learning (ICRL) setting by introducing rewards instead of ground truth answers. The authors show that the naive approach of populating a context with all interactions does not work due to lack of exploration, and propose how to mitigate the problem. Through experiments on a classification datasets, authors show the ability to improve over one-shot classification, however the proposed method is still bounded by a supervised in-context learning approach.

**Strengths:**

- The technicalities in the paper are clearly described, the ideas are communicated clearly
- Extensive ablations on different enhancements of the naive method
- The proposed method is better than zero-shot classification

**Weaknesses:**

[W1] I believe the contribution of this method is quite limited. Though it performs better than zero-shot classification, in 3 of the total 5 benchmarks ICRL underperforms supervised in-context learning (ICL) in terms of speed of convergence and accuracy. To my mind, if the method was promising, we would see at least comparable performance on a relatively easy task of text classification.

[W2] As it seemed to me, switching the setting from supervised ICL to ICRL is partly motivated by the ability of transformers to learn from reward signal, rather than from ground truth labels. However, the proposed method of generating rewards is basically grounded on the truth labels (as authors mention on line 252). Thus, the reward is a byproduct of ground truth $y^*$ and the claims of transition from supervised ICL to ICRL look vague to me.

[W3] Authors show that negative rewards make the performance of the methods worse. This contradicts the existing knowledge from ICRL research [1, 2]. In some of the recent works [3, 4] it is shown that transformers are able to improve from suboptimal demonstrations in-context. In essence, transformers are Bayesian learners [5, 2] and the more diverse a prior is, the better transformers can approximate posterior. This can be a signal that LLMs do not have an ability to maximize the reward in-context, which is a crucial feature of ICRL model trained from scratch [1, 2, 3, 4].

[1] Laskin, Michael, et al. "In-context reinforcement learning with algorithm distillation.”

[2] Lee, Jonathan, et al. "Supervised pretraining can learn in-context reinforcement learning.”

[3] Dai, Zhenwen, Federico Tomasi, and Sina Ghiassian. "In-context Exploration-Exploitation for Reinforcement Learning.”

[4] Zisman, Ilya, et al. "Emergence of In-Context Reinforcement Learning from Noise Distillation.”

[5] Müller, Samuel, et al. "Transformers can do bayesian inference.”

**Questions:**

1. Following [W1] and [W2], can authors, please, explain the motivation in more details? Why do we might want to turn the supervised ICL classification into the RL task? What is the difference between two approaches, if the reward signal is computed solely based on the ground truth. In what situation can we benefit from turning ICL into ICRL?
2. I believe, it would be more interesting to explore the abilities of LLMs on the RL tasks, rather than classification. To start with, authors can show LLMs can solve simple contextual bandits and MDP environments from [2], if the observations are prompted in a correct fashion. This would be a more obvious setting, since the reward would not be generated through ground truth labels, rather given from the environment once it is solved.

Although my initial recommendation is to reject the paper, I would love to actively participate in the discussion and to revise my score based on the authors’ response.

---

### Official Review · Reviewer_7AfD · 2024-11-04

**Soundness:** 3
**Presentation:** 3
**Contribution:** 2
**Rating:** 6
**Confidence:** 3

**Summary:**

The paper investigates in-context reinforcement learning (ICRL). In contrast to in-context learning (ICL), in ICRL, an LLM gets as inputs triplets: query, answer, evaluation. The authors propose some methods (e.g., add more randomization, use only positve evaluation examples) to make ICRL work.

**Strengths:**

The paper studies an interesting and promising research direction.

The authors identify two causes (i.e., lack of exploration and inability of learning from negative examples) for why a naive approach would fail. These observations may help future research in designing more efficient ICRL methods.

The paper is generally well-written and clear, although there are a few typos, e.g.:
Line 72: missing point
Line 186: deploy
Line 187: Similar to -> Similarly to
Line 426: too high -> too small?
Line 539: our

**Weaknesses:**

In contrast to what the title may suggest, the setting that is actually considered seems to be very restrictive, e.g., only binary evaluations, state-less reinforcement learning setting. This setting (and the actual proposition of only using positive triplets) makes it very similar to ICL. The main difference as far as I can see is that the LLM needs to explore to find good examples. Is my understanding correct?

If my understanding is correct, then the proposed method seems basically to perform random search to find correct examples and then follow ICL, which makes the proposed method (explorative ICRL) quite basic and straightforward.

**Questions:**

Is my understanding of the investigated ICRL setting correct?

---

### Official Review · Reviewer_hB8B · 2024-11-05

**Soundness:** 2
**Presentation:** 3
**Contribution:** 2
**Rating:** 3
**Confidence:** 4

**Summary:**

This paper proposes an in-context reinforcement learning (ICRL) framework for LLMs where tuples of states, predictions, and rewards are sent as input to the prompt. The authors observe that a naive implementation of ICRL fails due to a lack of exploration. To address this, they introduce two methods: 1. Explorative ICRL, which introduces randomness by modifying the prompt structure to encourage exploration, and 2. Approximate ICRL, which aims to overcome the computational costs.

**Strengths:**

1. This paper proposes 3 variants of in-context reinforcement learning to address issues such as exploration and computation cost.
2. The authors provide empirical evidence on how to achieve effective ICRL, highlighting that negative rewards are not useful in this context and that naive ICRL fails. They introduce approximate ICRL to mitigate computational costs, offering practical solutions.
3. The angle of utilizing LLM's stochatisticy for better ICL is interesting.
4. This paper is clearly written and organized.

**Weaknesses:**

1. The paper's approach to reinforcement learning is limited by using only positive rewards and filtering out negative ones. In traditional RL, learning from negative feedback is essential for policy improvement.
2. Prior works on in-context learning also have considered cases with only positive labels, raising questions about how this approach differs from standard few-shot prompting where the correct output y is always provided for every x.
3. The tasks considered in experiments are mostly text classification tasks and they only provide binary rewards to the agent. Experimenting with more complex text-based RL environments will be more convincing.
4. Lack of comparison with RL baselines. A simple baseline would be using RLHF on the binary rewards.
5. Since the author claimed stochasticity brings to exploration, a natural baseline is sampling with higher temperature, or simply reordering the context examples. This simple baselines are missing in the discussion to make this work more comprehensive.

**Questions:**

1. Following weakness 1: Since you only give positive rewards in the context, what would happen if you don't include the rewards in the prompt and treat it as few-shot demonstration. Would it achieve similar performance?

---

### Author Response · Authors · 2024-11-22

We thank the reviewers for their time and work. Given the misconceptions in the reviews and the low scores, we estimate little chance of a positive outcome from a discussion, so to respect everyone’s time, we are withdrawing the paper.

**However, because this forum will become public, we would like to clarify some important misconceptions:**

# How does ICRL differ from ICL? The ICL numbers are better, so what good is ICRL?

Reviewers noted that the results we showed for ICRL are worse than ICL, so has no utility. This comparison is moot. ICL is a supervised approach and requires annotated labels. ICRL relies on rewards for model predictions – it is simply a completely different learning scenario and one that makes a much weaker assumption on data annotation. We use labeled data to conduct experiments only (i.e., to compute rewards), and in domains with reward functions labels are not needed. This is a common practice when conducting RL experiments. We provide ICL curves to provide a broader understanding of our results. Interpreting these curves as comparable is wrong.

# Why focus on text classification only? Is this RL?

More complex domains are an important direction for future work. Text classification allows for very clean and relatively scalable experiments. It’s also relatively well understood how to encode such tasks in context. Other domains have pros and cons, complicating the experimental setup, but allowing to stress test aspects of RL not well covered by text classification. Our use of text classification does not undermine the clear and significant answers we find to the research questions we study.

Classification creates a bandit setting, which is a special case of RL. We do not claim that every RL problem will work with LLMs. But, simply, that with the right process, LLMs **can** show RL behavior.

# Why should we care about in-context reinforcement learning?

The in-context learning ability of LLMs has been the focus of significant research in recent years. Our study is a further exploration of these abilities. So, in brief, it seems that many people care about in-context learning abilities, and our work is clearly within this scope. Establishing a good understanding of RL abilities in context also has applied implications, opening the way for using in-context learning without labels (but instead with reward signals). Our work is a starting point, showing for the first time this ability is possible (concurrent work from Google shows such abilities, but with fine-tuning).

# Does the use of examples with positive rewards nullify the entire approach?

No! First of all, we experiment with both positive- and negative-reward examples. We empirically show that LLMs don’t reason well about negative-reward examples, so our best performing results only use positive-reward examples. This is a data-driven empirical decision. It exposes an important direction for future work. But does it make our approach less RL? No! 0-1 binary reward functions are common in the field, and with policy gradient this basically means there are no negative-reward examples.

---

### Note · Authors · 2024-11-22

I have read and agree with the venue's withdrawal policy on behalf of myself and my co-authors.